# Bacteriophages isolated from mouse feces attenuates pneumonia mice caused by *Pseudomonas aeruginosa*

Nuttawut Sutnu[1,2‡], Wiwat Chancharoenthana[3,4‡]*, Supitcha Kamolratanakul[3,4], Pornpimol Phuengmaung[1,2], Uthaibhorn Singkham-In[1,2,5], Chiratchaya Chongrak[1,2], Sirikan Montathip[1,2], Dhammika Leshan Wannigama[1,6,7,8,9], Tanittha Chatsuwan[1], Puey Ounjai[10], Marcus J. Schultz[11,12,13], Asada Leelahavanichkul[1,2]*

1 Department of Microbiology, Faculty of Medicine, Chulalongkorn University, Bangkok, Thailand,
2 Department of Microbiology, Center of Excellence in Translational Research in Inflammation and Immunology (CETRII), Faculty of Medicine, Chulalongkorn University, Bangkok, Thailand, 3 Department of Clinical Tropical Medicine, Faculty of Tropical Medicine, Mahidol University, Bangkok, Thailand,
4 Department of Clinical Tropical Medicine, Tropical Immunology and Translational Research Unit (TITRU), Faculty of Tropical Medicine, Mahidol University, Bangkok, Thailand, 5 Faculty of Medical Technology, Rangsit University, Pathum Thani, Thailand, 6 Department of Infectious Diseases and Infection Control, Yamakata Prefectural Central Hospital, Yamakata, Japan, 7 Department of Infectious Diseases and Infection Control, Pathogen Hunter's Research Collaborative Team, Yamakata Prefectural Central Hospital, Yamakata, Japan, 8 School of Medicine, Faculty of Health and Medical Sciences, The University of Western Australia, Perth, WA, Australia, 9 Biofilms and Antimicrobial Resistance Consortium of ODA Receiving Countries, The University of Sheffield, Sheffield, United Kingdom, 10 Department of Biology, Faculty of Science, Mahidol University, Bangkok, Thailand, 11 Department of Intensive Care & Laboratory of Experimental Intensive Care and Anesthesiology (L.E.I.C.A), Academic Medical Center, University of Amsterdam, Amsterdam, Netherlands, 12 Mahidol-Oxford Tropical Medicine Research Unit (MORU), Mahidol University, Bangkok, Thailand, 13 Nuffield Department of Medicine, Centre for Tropical Medicine and Global Health, Oxford University, Oxford, United Kingdom

‡ NS and WC are co-first authors of this work.
* wiwat.cha@mahidol.ac.th (WC); aleelahavanit@gmail.com (AL)

**Data Availability Statement:** All relevant data are within the manuscript and its Supporting Information files.

## Abstract

### Background

Most of the current bacteriophages (phages) are mostly isolated from environments. However, phages isolated from feces might be more specific to the bacteria that are harmful to the host. Meanwhile, some phages from the environment might affect non-pathogenic bacteria for the host.

### Methods

Here, bacteriophages isolated from mouse feces were intratracheally (IT) or intravenously (IV) administered in pneumonia mice caused by *Pseudomonas aeruginosa* at 2 hours post-intratracheal bacterial administration. As such, the mice with phage treatment, using either IT or IV administration, demonstrated less severe pneumonia as indicated by mortality, serum cytokines, bacteremia, bacterial abundance in bronchoalveolar lavage fluid (BALF), and neutrophil extracellular traps (NETs) in lung tissue (immunofluorescence of neutrophil elastase and myeloperoxidase).

**Funding:** This research project is supported by the NSRF via the Program Management Unit for Human Resources & Institutional Development, Research and Innovation (B16F640175) and (B36G660003) and (B48G660112), Rachadapisek Sompote Matching Fund (RA-MF-22/65 and RA-MF-13/66), the National Research Council of Thailand (NRCT-N41A640076 and NRCT-N34A660583), and Fundamental Fund 2567. NS is funded by the 90th Anniversary of Chulalongkorn University Fund (Ratchadapisek Sompote Endowment Fund). WC is funded by Mahidol University (Fundamental Fund; Basic Research Fund: fiscal year 2022).

**Competing interests:** The authors have declared that no competing interests exist.

## Results

Interestingly, the abundance of phages in BALF from the IT and IV injections was similar, supporting a flexible route of phage administration. With the incubation of bacteria with neutrophils, the presence of bacteriophages significantly improved bactericidal activity, but not NETs formation, with the elevated supernatant IL-6 and TNF-α, but not IL-1β. In conclusion, our findings suggest that bacteriophages against *Pseudomonas aeruginosa* can be discovered from feces of the host.

## Conclusions

The phages attenuate pneumonia partly through an enhanced neutrophil bactericidal activity, but not via inducing NETs formation. The isolation of phages from the infected hosts themselves might be practically useful for future treatment. More studies are warranted.

## Introduction

Pneumonia is an inflammatory response against infections from several organisms that primarily affect the alveoli and the distal bronchial tree of the lungs. Pneumonia from bacterial infection is the most common condition, especially in the elderly. The incidence of severe pneumonia in the elderly ($> 65$ years old) is approximately 10–13 per 10,000 population [1]. Although *Streptococcus pneumoniae*, *Haemophilus influenzae*, and *atypical bacteria* (*Chlamydia pneumoniae*, *Mycoplasma pneumoniae*, and *Legionella species*) are the most common causes of community-acquired pneumonia [2], pneumonia from Gram-negative bacteria (mostly *Pseudomonas aeruginosa* and *Acinetobacter baumannii*) are important causes of ventilator-associated pneumonia [3]. Interestingly, the mortality rate of pneumonia from Gram-negative bacteria is higher than Gram-positive bacteria and *P. aeruginosa* is the most common cause of hospital-acquired pneumonia and healthcare-associated pneumonia [4]. Additionally, *P. aeruginosa* is an opportunistic pathogen with an increasing incidence of antibiotic resistance [5, 6] and bacteriophage therapy is an interesting adjunctive treatment.

As such, bacteriophages (phages) are viruses that infect bacteria through specific receptors that are categorized into 2 major patterns, including lytic and lysogenic cycles [7]. For lytic cycle, phages produce new viral progenies in the bacterial host using host materials for viral DNA replication, synthesis, and phage assembly before releasing new phage particles through disrupted bacterial hosts [8]. In the lysogenic mechanism, phages integrate their genomes into the bacterial chromosome and coexist within the host, referred to as "prophages", and some prophages can differentiate to the lytic cycle under stressful conditions [9, 10]. Nevertheless, the extremely high specificity of the restricted bacterial receptors results in the limitation of the use of phages against bacterial infection [11]. For example, phages that positively affect *P. aeruginosa* in one host might not have an effect on *P. aeruginosa*, the same species, in another host. Indeed, bacteriophages are highly species-specific; often, they infect only one species of bacteria or even particular strains within a species [11]. Hence, a library of bacteriophages and combined phages against a single bacterial species are necessary to enhance the possibility that some phages from the library can cover this particular bacterial strain. Currently, phages are mostly isolated from wastewater or other environmental sources partly due to the high abundance of phages in these places [12]. However, some phages from the environment infect only non-harmful bacteria toward the hosts, while phages derived from a living animal could also

be an interesting and safe source of phages against microbes infecting the host. We chose mouse feces for the proof of concept because i) humans are mammals and phages isolated from mammals may be more similar to their use in humans, and ii) experimental mice are small enough to use as an initial recovery experiment and future experiments for improved phage isolation in mice are feasible and easier than the larger mammals.

Although the treatment using phages isolated from the host to treat that host is an interesting idea, data on phage isolation from the living host are still less. In mammals, most bacteriophages can be isolated from the areas with bacterial colonization, including the intestine, urogenital tract, upper respiratory tract, and skin, where the highest phage abundance is in the gut [13] and feces might be an interesting source of phages. Since phage isolation doesn't necessitate a significant amount of feces, fecal collection and other manipulations on mice are more straightforward than in larger animals; for instance, administering certain molecules that could potentially enhance the target phage. Additionally, phages targeting specific bacteria may exhibit greater specificity in certain mouse strains that have undergone genetic modifications, a prospect for future research. Moreover, phages also enhanced bactericidal activities of some immune cells, especially neutrophils [14], and immune cells are necessary for phage effectiveness [15–18], especially against the multidrug-resistant pathogens [19–21]. Although the idea of using phages that are isolated from the host to treat the individual host without the necessity of a phage library is interesting, successful phage isolation from the host is needed as an initial step. Additionally, the effectiveness of the treatment with bacteriophages in *Pseudomonas* pneumonia and the proper routes for phage administration in pneumonia are still uncertain. Therefore, we suggest that bacteriophage therapy could serve as an alternative therapy or an adjuvant strategy to combat bacterial infections. Hence, this study was designed to evaluate the possibility of isolating phages against *P. aeruginosa* from mouse feces preparing for the possible use of phages isolated from the host in the next step.

## Methods

### Bacterial isolates and culture conditions

The clinically isolated organisms from the human sample in the central laboratory in the King Chulalongkorn Memorial Hospital of the Department of Microbiology, Faculty of Medicine, Chulalongkorn University under the institutional review board (IRB) number 0285/65 from the Faculty of Medicine, Chulalongkorn University according to the Declaration of Helsinki. The human records were not needed for the study and all data (source of the organisms) were fully anonymized and the ethics committee waived the requirement for informed consent. Notably, on June 1, 2022, the IRB approved the collection of isolated bacteria, and on December 30, 2022, the study came to an end due to the collection of an adequate number of bacterial strains. As such, the isolated stains of all currently available bacteria, including *P. aeruginosa* (PA1, PA2, PA3, PA4, PA5, PA6, PA7, PA8, PA9, PA10, PA11, and PACL), *Klebsiella pneumonia* (KP1, KP2, KP3, KP4, and KP5) *Staphylococcus aureus* (SA1, SA2, and SA3), and *Enterococcus faecalis* (EF1, EF2, and EF3) were isolated from patients of the King Chulalongkorn Memorial Hospital, Bangkok, Thailand. The commercially available bacteria, including *P. aeruginosa* ATCC 27853 and PAO1, *Escherichia coli* ATCC 25922, and *S. aureus* ATCC 29213 were also used. All isolates were grown in trypticase soy broth (TSB; Difco™, Becton, NJ, USA) and incubated at 37°C, 200 rpm for 18 h.

### Bacteriophage isolation and purification

Bacteriophage isolation was performed according to a published protocol [13]. Briefly, mouse feces (from the cecum and colon) were mixed with PACL (used as hosts for phages) and

incubated at 37˚C for 18 h before centrifugation at 5,000 ×$g$, 4˚C for 30 min to separate the supernatant before passing through a 0.22 μm filter. The screening of phages against PACL was performed by plaque assay [22]. In brief, overnight culture of PACL was transferred to soft agar, Tryptic Soy Broth (TSB) with 0.7% agar (on the top part) mixed with filtered bacteriophage supernatant and overlaid on the Tryptic Soy Agar (TSA; at the bottom part) and incubated at 37˚C for 16 h to allow plaques (bacteriophages) forming. A single plaque was transferred, dissolved in 1X SM buffer (150 mM NaCl, 50 mM Tris-HCl pH7.5, 10 mM MgSO$_4$, and 1 mM CaCl$_2$; pH 7.5), and amplified using the same method as described above. Then, phage purification with the cesium chloride (CsCl; Merck) density gradient method, the DNA separation from the sample containing DNA, RNA, and proteins using the property of the same density between CsCl and DNA, was performed following a previous publication [22]. Briefly, the gradient of CsCl (Merck, Germany) at 1.6 g/cm$^3$, 1.5 g/cm$^3$, and 1.3 g/cm$^3$ was added into an ultracentrifuge tube at the ratio of 2:3:3, while crude bacteriophage suspensions were fully added onto the top of layer in a tube. The CsCl density gradients separated the purified phages by centrifugation at 80,000 ×$g$, 4˚C for 3 h. After centrifugation, the visible band containing the bacteriophage was isolated by puncturing the thin-walled ultracentrifuge tube with a 26-gauge needle and syringe. The purified bacteriophage was dialyzed with 100-kDa MWCO dialysis tubing in SM buffer at 4˚C for 24 h to remove the CsCl. The bacteriophage was stored at -80˚C in 7% dimethyl sulfoxide (DMSO; Merck) for further analysis (**S1 Fig**). For morphological analysis, the purified phages were dropped into grids containing carbon film 200 mesh copper (Electron Microscopy Sciences, PA, USA), negatively stained with 2% uranyl acetate, dried with filter paper, and visualized by a transmission electron microscope (JEM 1400 Plus, JOEL, MA, USA) at an accelerating voltage of 80 kV.

## Host range determination, bactericidal activity, and one-step growth curve

The host range by spot test on 27 different bacterial isolates was conducted using 3 mL of 0.7% agar TSB mixed with 1 mL of each bacterial culture on TSA using 4 isolated bacteriophages from feces of healthy mice (Table 1). After solidification of the top agar, 10 μL of bacteriophage (~$10^9$ PFU/mL) and 10-fold serially diluted in SM buffer were spotted on the plate and incubated for 18 h at 37˚C to detect the presence of bacterial lysis. For bactericidal activity, overnight culture of PACL ($1 \times 10^6$ CFU/mL) in 96-well plates (200 μL/well) with or without bacteriophage suspension at multiplicity of infection (MOI) of 0.001, 0.01, 0.1, 1, and 10, were incubated at 37˚C before bacterial enumeration (bacterial turbidity at 600 nm of spectrophotometer) every 1 h intervals for 24 h. For the one-step growth curve, PACL was mixed with bacteriophages at MOI 0.01 for 10 min at 37˚C with 200 rpm shaking before centrifugation at 12,000 rpm for 2 min. The pellet was suspended with 5 mL TSB and incubated at 37˚C and 200 rpm. During the incubation, a sample (200 μL) was taken every 10 min between 0 and 160 min before estimating phage quantity by plaque assay.

## Animal and animal model

According to the National Institutes of Health (NIH) criteria, the Institutional Animal Care and Use Committee of Chulalongkorn University, Bangkok, Thailand approved the animal protocol (007/2566), and the experiments began after 1 July 2023 because the animal study protocol's approval date was 1 June 2023.

Male 12-week-old C57BL/6 mice were intratracheally (IT) administered with 0.2 mL of $10^8$ CFU/mL of PACL or normal saline solution before being treated with bacteriophage (0.2 mL of $10^9$ PFU/mL) or NSS (negative control) via the intravenous (IV; tail vein injection) or IT route at 2 h after bacterial inoculation. Mice were observed for 7 days and 24 h for survival

**Table 1. Host range determination of Pseudomonas phages.**

| Species | ID (vB_PaeM) | AL | AM | AN | AO |
|---|---|---|---|---|---|
| *P. aeruginosa* | PA1 | - | - | - | - |
| *P. aeruginosa* | PA2 | + | + | - | - |
| *P. aeruginosa* | PA3 | + | - | - | - |
| *P. aeruginosa* | PA4 | + | - | + | - |
| *P. aeruginosa* | PA5 | + | - | - | - |
| *P. aeruginosa* | PA6 | + | + | - | - |
| *P. aeruginosa* | PA7 | + | - | - | + |
| *P. aeruginosa* | PA8 | - | - | - | - |
| *P. aeruginosa* | PA9 | + | + | - | + |
| *P. aeruginosa* | PA10 | - | - | - | - |
| *P. aeruginosa* | PA11 | + | - | + | - |
| *P. aeruginosa* | PA12 | - | - | - | - |
| *P. aeruginosa* | PA13 | + | - | - | - |
| *P. aeruginosa* | PA14 | - | - | - | - |
| *P. aeruginosa* | PACL | + | - | - | - |
| *P. aeruginosa* | ATCC 29213 | - | - | - | - |
| *P. aeruginosa* | PAO1 | - | - | - | - |
| *K. pneumonia* | KP1 | - | - | - | - |
| *K. pneumonia* | KP2 | - | - | - | - |
| *K. pneumonia* | KP3 | - | - | - | - |
| *K. pneumonia* | KP4 | - | - | - | - |
| *K. pneumonia* | KP5 | - | - | - | - |
| *S. aureus* | SA1 | - | - | - | - |
| *S. aureus* | SA2 | - | - | - | - |
| *S. aureus* | SA3 | - | - | - | - |
| *S. aureus* | ATCC 29213 | - | - | - | - |
| *E. faecalis* | EF1 | - | - | - | - |
| *E. faecalis* | EF2 | - | - | - | - |
| *E. faecalis* | EF3 | - | - | - | - |
| *E. coli* | ATCC 25922 | - | - | - | - |

+ Present plaque

No plaque

analysis and short-term parameters, respectively, before sacrifice by cardiac puncture under isoflurane anesthesia with sample collection, including blood, bronchoalveolar lavage fluid (BALF), and lung tissue. The abundance of bacteria and bacteriophages was determined by colony counting and plaque assay, respectively. Serum cytokines (TNF-$\alpha$, IL-6, IL-1$\beta$, and IL-10) were measured using enzyme-linked immunosorbent assay (ELISA, Invitrogen). There were 7 mice per group and 7 groups (total 49 mice) were used for survival experiments, while another 49 mice (7 groups) were used for determination of 24 h parameters. The model was induced by well-trained investigators with certificates and mice were observed every 3 h for 9 h post induction and then at 18 and 24 h of the model. The animal welfare was considered and humane endpoint for the moribund mice was applied using the Murine Sepsis Score (MSS score) based on appearance (piloerection), responses to stimuli (consciousness and activity), eye opening, and respiratory quality (laboured breathing) with the score 0 to 3 for each parameter [23]. In our experiment, there was no found dead animal and the most common

indication for sacrifice was labored breathing (mostly at 24 h of pneumonia induction). The mice with the score higher than 4 were humanely euthanized immediately at the detected time of the observation. The sacrifice method used was cardiac puncture under isoflurane anesthesia before collection of mouse samples.

### Histological analysis

Lungs, fixed with 10% neutral buffered formalin, were stained with hematoxylin and eosin (H&E) color in 5 μm thickness slides before semi-quantitatively evaluated (200× magnification in 10 randomly selected fields) for desquamation, dystelectasis/atelectasis, congestion, interstitial thickness, infiltration, and bronchial exudate with the following score; 0 indicated no injury; 1 indicated minimal or discrete injury; 2 indicated mild injury; 3 indicated moderate injury; and 4 indicated severe injury [24]. In parallel, neutrophil extracellular traps (NETs) in the lung were determined by immunofluorescence analysis using embedded tissue in Tissue-Tek O.C.T Compound stained with antibodies targeting neutrophil elastase (NE), myeloperoxidase (MPO), and citrullinated histone H3 (CitH3) with 4′,6-diamidino-2-phenylindole (DAPI) for nuclear visualization. The fluorescent images and areas with fluorescence were interpreted using a ZEISS LSM 800 Airyscan confocal laser scanning microscope at ×630 magnifications (Carl Zeiss, Germany).

### In vitro experiments on neutrophils

Mouse neutrophils were extracted at 3 h after 3% thioglycolate intraperitoneal injection (1 mL) as previously described [25, 26]. The samples were collected, washed with ice-cold phosphate buffer solution (PBS), and centrifuged at 1,800 rpm, 4°C for 5 min, resuspended with fresh Roswell Park Memorial Institute media (RPMI) (Gibco-Invitrogen, Grand Island, NY, USA) supplemented with 10% heat-inactivated FBS (Gibco-Invitrogen). Then, $5 \times 10^5$ neutrophils were seeded on cover glass coated with poly-L-lysine (Sigma-Aldrich) in 24-well plates and treated with bacteriophages ($10^7$ PFU/mL), PACL ($10^6$ CFU/mL), or PACL with bacteriophages for 2 h at 37°C in a 5% $CO_2$ incubator. Subsequently, the coverslips were processed for visualization of NETs using antibodies against NE, MPO, and CitH3, with DAPI staining [27, 28]. Additionally, bacterial and bacteriophage counts from supernatants using colony counting and plaque assay, respectively, with supernatant cytokine measurement by ELISA (Invitrogen) were also performed.

### Statistical analysis

Statistical differences were examined using an unpaired student's *t*-test or one-way analysis of variance (ANOVA) with Tukey's post hoc test to analyze two or multiple groups, respectively. Survival analysis was determined by the Log-rank test. Data were presented as mean ± standard error with $p < 0.05$ considered significant using GraphPad Prism version 10.2.0 (GraphPad Software, Inc., San Diego, CA, USA).

## Results

### Isolation and characterization of a bacteriophage from mouse feces

Different from the isolation from environmental sources, there were only 4 different bacteriophages from our results (**Table 1**) and pseudomonas phage vB_PaeM-AL was selected for further test in vivo and in vitro due to the best coverage in most of *P. aeruginosa* strains used in our experiments. As such, pseudomonas phage vB_PaeM-AL, isolated from feces of healthy mice, showed effective lytic activity against 10 from 17 strains of *P. aeruginosa* isolates

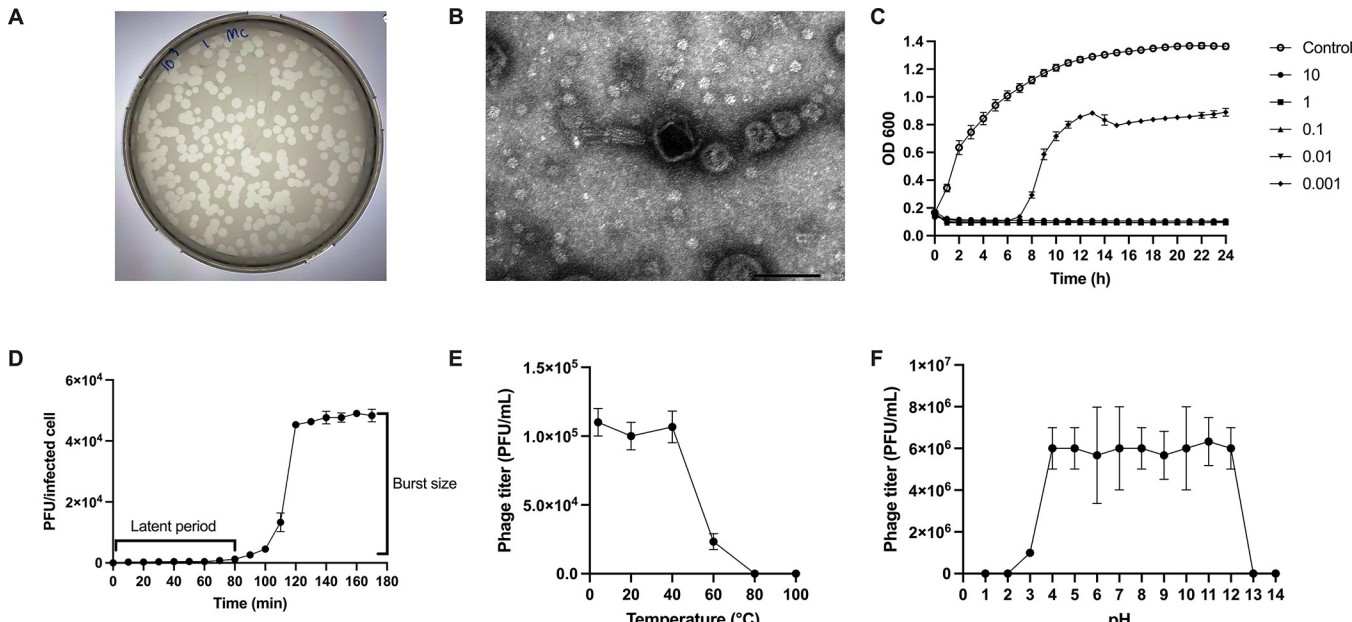

**Fig 1.** Characterization of Pseudomonas phage vB_PaeM-AL as indicated by a representative pictures of plaque morphology on double agar layer plate and transmission electron micrograph (**A, B**), bactericidal activity (**C**), one-step growth curves (**D**), and effects of temperature and pH (**E, F**) on lytic activity are demonstrated. The results were combined from the isolated triplicated experiments. Data are shown as the mean ± SEM.

(58.82%) without an impact on other bacteria, while other isolated phages covered only a few pseudomonas strains (**Table 1**). Clear plaques with approximately 2 mm diameter were demonstrated (**Fig 1A**). Based on morphology, this phage belongs to the Myoviridae family, which is characterized by an isometric capsid and a relatively short contractile tail. The size of the capsid was approximately 73 nm and the contractile tail measured approximately 65 and 26 nm in length and width, respectively (**Fig 1B**). For bactericidal activity, the optical density of the uninfected cells without bacteriophages increased continuously during the incubation, while bacteria with phages at MOIs of 0.01, 0.1, 1, and 10 showed a reduction in bacterial growth throughout the entire incubation duration (**Fig 1C**). However, there was a decrease in bacterial growth initially, followed by an increase after 7 hours of incubation at the MOI of 0.001 (**Fig 1C**). The latency period (the duration from phage infection to the lysis host cells) was approximately 80 min with the burst size at 188 PFU/infected cell (**Fig 1D**). With 60 min incubation, the phages maintained stable infectivity after incubation at 4, 20, and 40˚C, while decreased at 60, 80, and 100˚C (**Fig 1E**). At 60 min of incubation, the bacteriophage exhibited stable infectivity across a broad range of pH, specifically from pH 3 to pH 13 (**Fig 1F**).

## Impacts of Pseudomonas phage vB_PaeM-AL on *Pseudomonas* pneumonia mice

Although Pseudomonas phage vB_PaeM-AL can neutralize several pseudomonas strains (**Table 1**), PACL, a strain with previously demonstrated pathogenicity in an animal model (wound model) [6], might also be able to induce mouse pneumonia was selected to use for further experiments. Because the minimal lethal dose (MLD) that triggered 100% bacterial death within 7 days was $5 \times 10^7$ CFU, mice were intratracheally (IT) challenged with $2 \times$ MLD ($10^8$ CFU) of *Pseudomonas* (PACL) with $10^9$ PFU of phages. Phage treatment, either through intratracheal (IT) or intravenous (IV) routes, attenuated pneumonia as indicated by survival

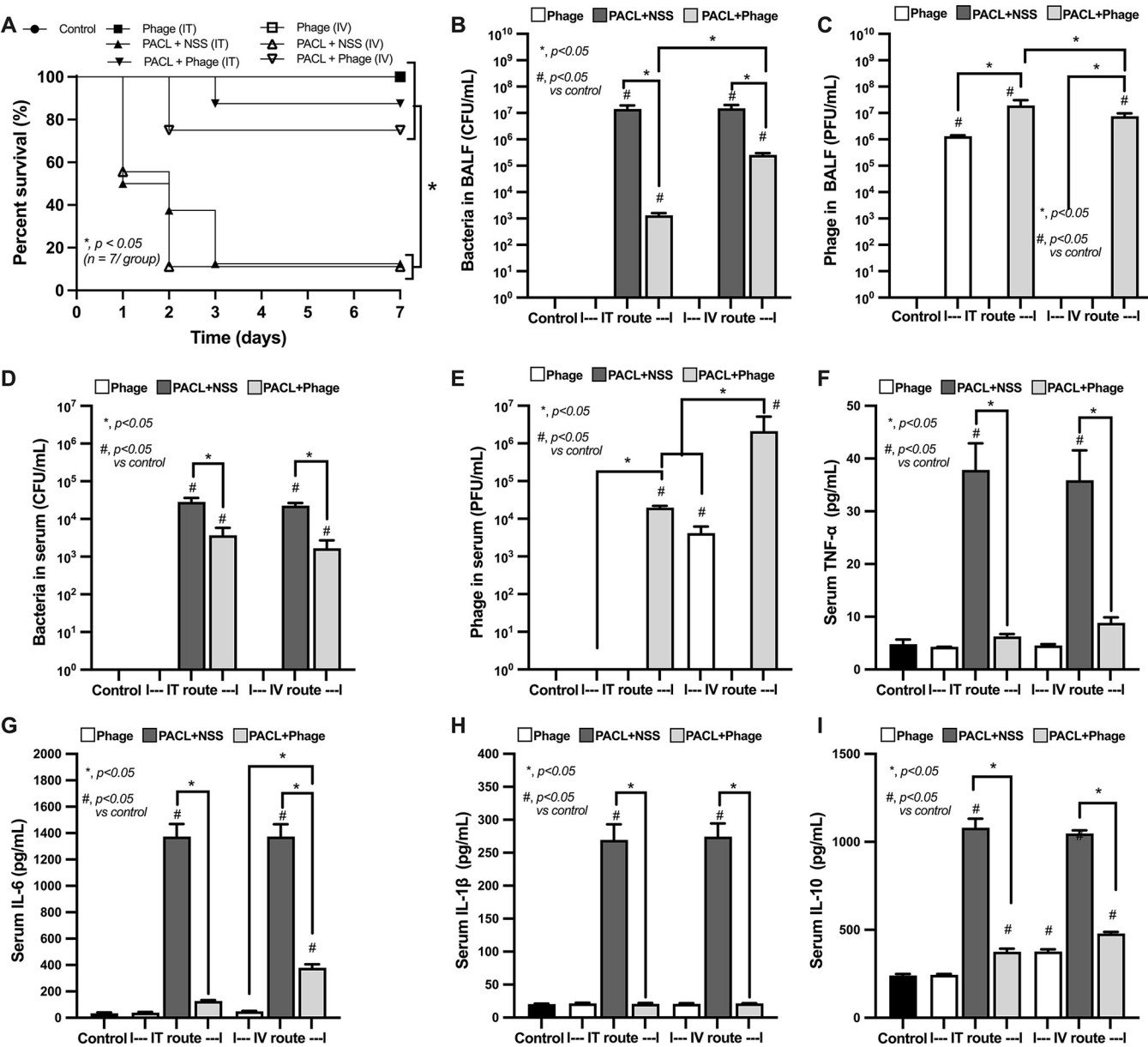

**Fig 2.** Characterization of *Pseudomonas aeruginosa*-induce pneumonia mice with normal saline (NSS) or phage vB_PaeM-AL using intravenous (IV) and intratracheal (IT) routes as indicated by survival analysis (**A**), bacterial loads and phage titer (**C**) in bronchoalveolar lavage fluid (BALF) and in blood (**B-E**), and serum cytokines (TNF-α, IL-6, IL-1β, and IL-10) (**F-I**) are demonstrated (n = 7/ group). Data are shown as the mean ± SEM.

analysis, bacterial abundance in blood and in bronchoalveolar lavage fluid (BALF), and serum cytokines with similar phage abundance between IT and IV injection at 24 h of the model (**Fig 2A–2I**). Without phage treatment, the lethal pneumonia of our model was indicated by approximately 90% mortality rate within 7 days after bacterial administration (**Fig 2A**) with high bacterial abundance (BALF and in blood), elevated inflammatory cytokines (**Fig 2B–2I**), and lung consolidation (the presence of exudate in the airways and alveoli) (**Fig 3A and 3B**), which was compatible with severe lobar pneumonia in patients. With phage treatment, there was a higher abundance of phages in serum after IV injection than the IP administration (**Fig 2E**). Notably, there was no difference in survival rate between IT and IV phage administration

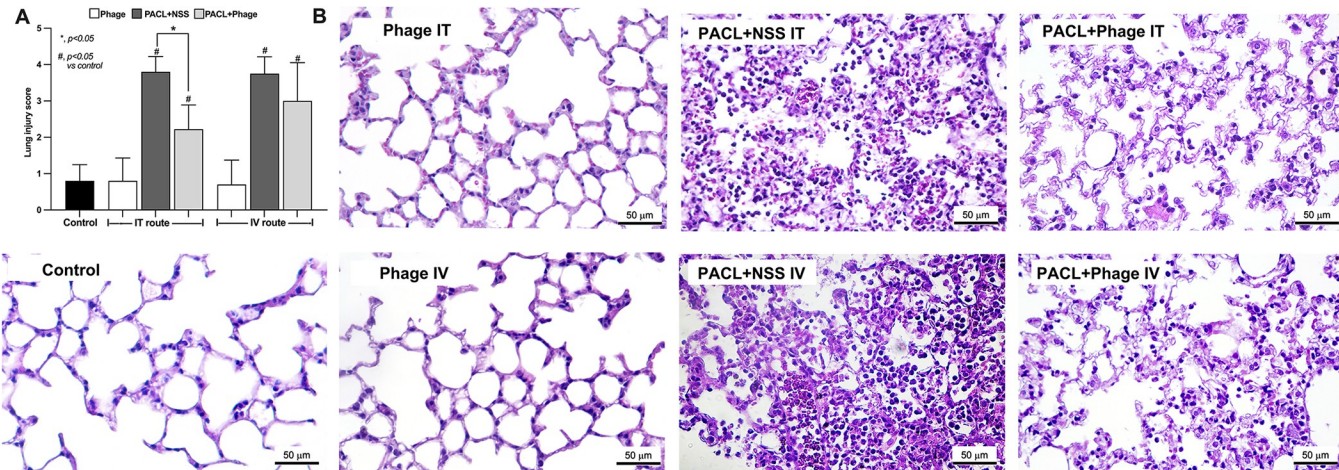

**Fig 3.** Histopathological examination of lung tissue harvested from mice with *Pseudomonas*-associated pneumonia treated with and without bacteriophage as indicated by the lung injury score (**A**) and representative hematoxylin and eosin color (H&E) staining pictures (**B**) are demonstrated (n = 7/ group). Data are shown as the mean ± SEM.

in pneumonia mice (**Fig 2A**). The IV phage administration exhibited a greater reduction in bacteremia than the IT administration, while there was a similar abundance of phage in BALF after IT versus IV routes (**Fig 2C and 2D**), implying an effective bactericidal activity with rapid blood-tissue transfer of phages. On the other hand, there was no significant difference in serum cytokines between IT and IV routes and the administration of phages alone in the healthy mice increased phages in blood without the alteration in other parameters (**Fig 2A–2F**). In parallel, lung histology was similarly improved by either IV or IT phage administration (**Fig 3A and 3B**). Likewise, both IT and IV phage treatment similarly attenuated neutrophil extracellular traps (NETs) in lung tissue as indicated by the staining with neutrophil elastase (NE), myeloperoxidase (MPO), and citrullinated histone 3 (CitH3) (**Fig 4A–4D**).

## Enhanced neutrophil bactericidal activity by Pseudomonas phage vB_PaeM-AL

To test neutrophil activities, *P. aeruginosa* was incubated with polymorphonuclear (PMN) cells with and without phages for 2 h (**Fig 5A–5F**). As such, bacterial abundance in vitro was significantly decreased in the samples with phages regardless of the presence of PMN (**Fig 5A**), implying the possible effectiveness of phage in bactericidal activity without the requirement of bactericidal effect from PMN. However, the abundance of phages in the group with bacteria with neutrophils was lower than the bacteria without neutrophils (**Fig 5B**), implying a possible phage destruction by PMN. Hence, the similar bactericidal activity between bacteria with phage and bacteria with phage plus PMN (**Fig 5A**) with the lower abundance of phage in the latter group (**Fig 5B**), suggesting that both PMN and phage additively elevate bactericidal activity. Indeed, in neutrophils with phages without bacteria, there was an elevation of supernatant IL-6, but not other cytokines (**Fig 5C–5F**), supporting the neutrophil reactions from phage recognition. With the presence of PMN, bacteria plus phage elevated TNF-α and IL-6, but lower IL-1β and non-different IL-10 (**Fig 5C–5F**), when compared with bacteria alone, supporting cytokine-enhanced bactericidal activity of neutrophils [29]. For NETs, bacteria alone more potently activated NETs than bacteria plus phages and phage alone did not induce NETs, as indicated by NE, MPO, and CitH3 staining (**Fig 6A-6D**), supporting the more prominent impact of bacteria than phages on NETs formation and the different NETosis pathways

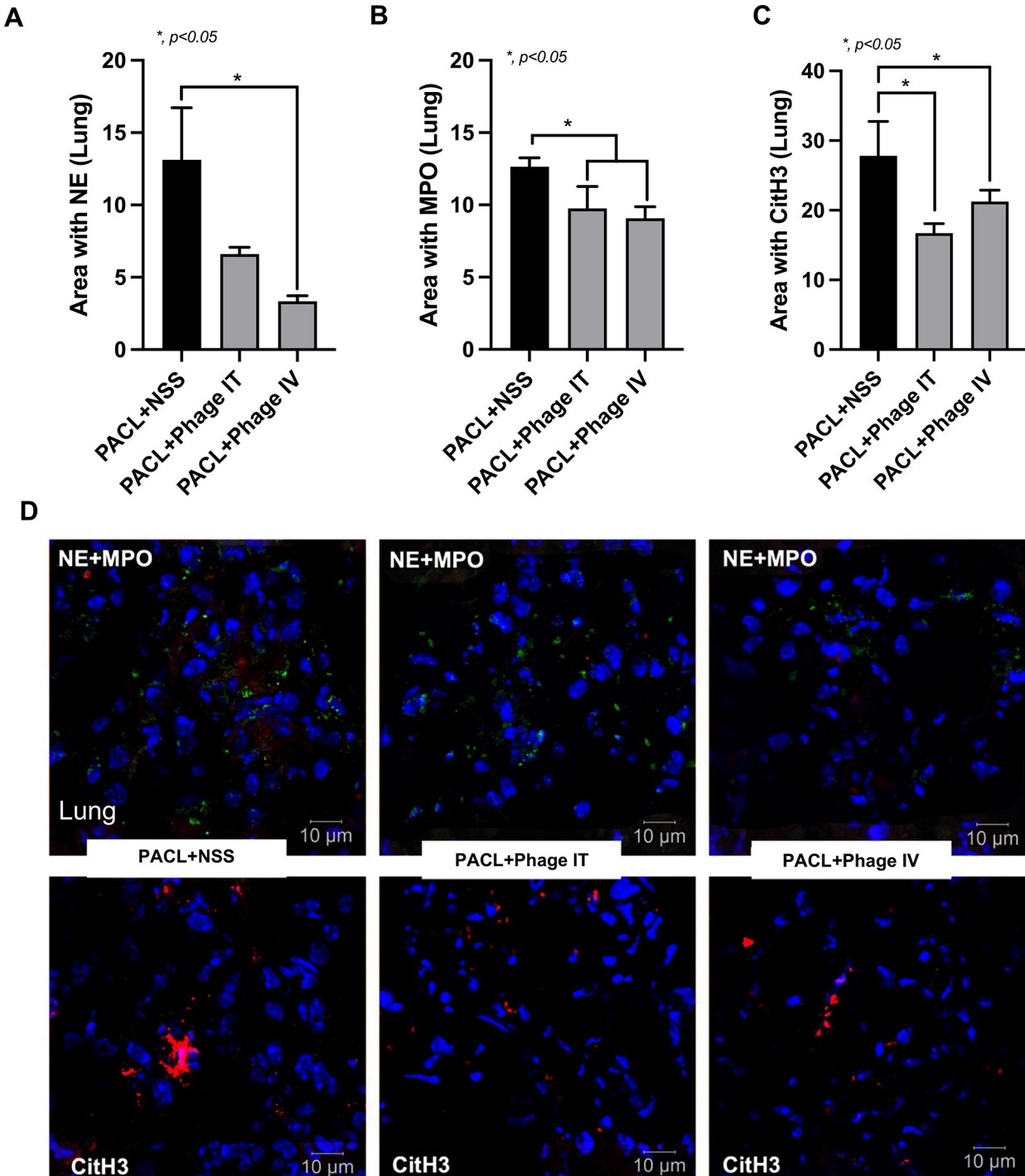

**Fig 4.** Neutrophil extracellular traps (NETs) in lung tissue harvested from mice with *Pseudomonas*-associated pneumonia after treatment with normal saline (NSS) or Pseudomonas phage vB_PaeM-AL using intravenous (IV) and intratracheal (IT) as indicated by immunofluorescence staining for neutrophil elastase (NE) (red), myeloperoxidase (MPO) (green), and citrullinated histone 3 (CitH3) (red), and presented by the area with NE (**A**), MPO (**B**), CitH3 (**C**) and the representative pictures (**D**) are demonstrated (n = 7/ group). Data are shown as the mean ± SEM.

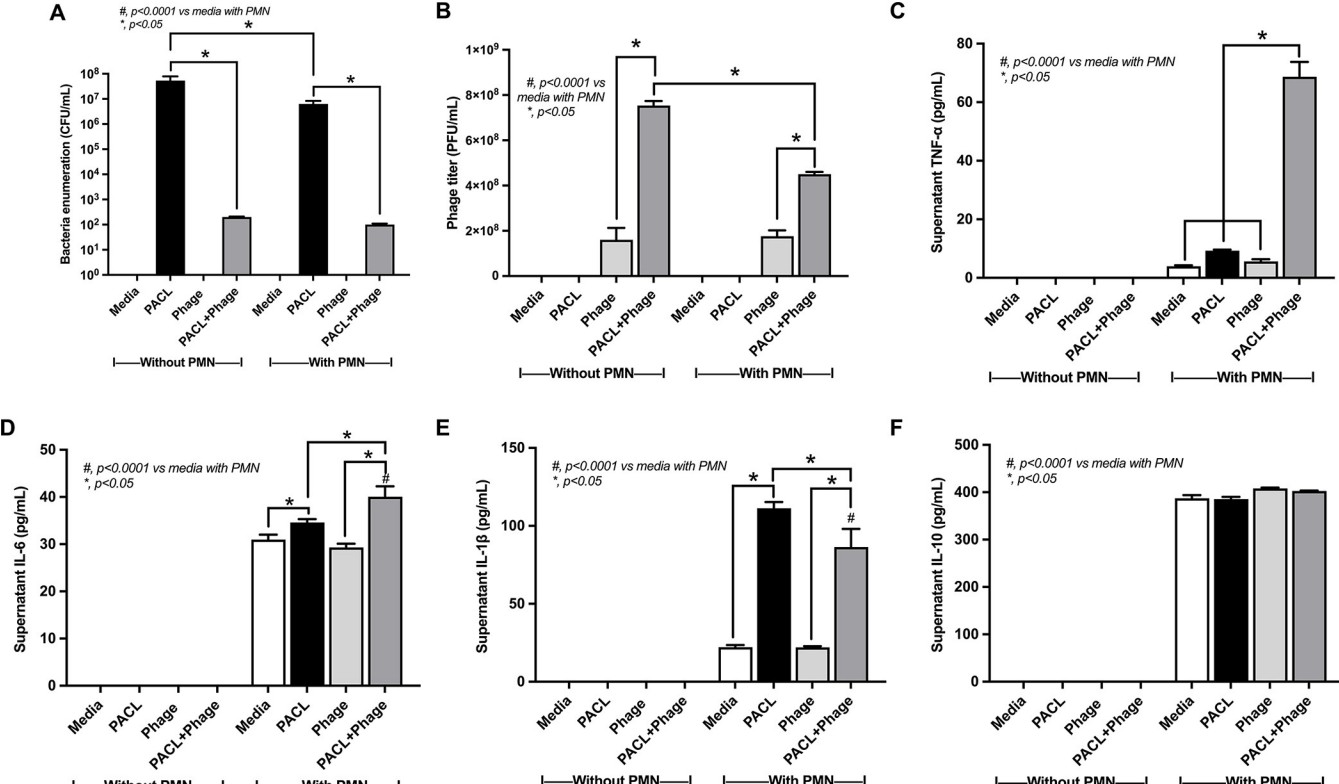

**Fig 5.** The clearance of *Pseudomonas aeruginosa* following the co-incubation with Pseudomonas phage vB_PaeM-AL with and without polymorphonuclear cells (PMN) as indicated by the abundance of bacteria and phages (**A, B**), and supernatant cytokines (TNF-α, IL-6, IL-1β, and IL-10) (**C-F**) are demonstrated. The results were combined from the isolated triplicated experiments. Data are shown as the mean ± SEM.

between bacterial and viral activations [30]. Although bacterial abundance between bacteria + phage + PMN and bacteria + phage was similar (**Fig 5A**)**,** lower phages in the latter group was required for this similar reaction (**Fig 5B**) and PMN could recognize phages as indicated by increased IL-6 (**Fig 5D** **on phage + PMN group**), but not NET activation (**Fig 6A-6D**)**.** Thus, bactericidal activity of our phage might, at least in part, be through the activation of neutrophils (cytokine production but not NETs).

## Discussion

Bacteriophage therapy is an interesting adjuvant or alternative strategy to antibiotics against multidrug-resistant bacteria [7]. Despite the usual source of bacteriophages from the wastewater [31], our lytic anti-pseudomonas phages were successfully isolated from the feces of the healthy mice supporting feces as an interesting source of bacteriophages. However, the sequencing analysis on the isolated phages was not performed because the discovery of new phages is not our primary objective. Here, the large burst size of Pseudomonas phage vB_PaeM-AL (188 PFU/infected cell at 80 min) implied the high level of virion replication leading to the high phage abundance as another beneficial bactericidal activity [32]. Likewise, the stable lytic activity at a wide range of pH (pH 4 to 10) and temperatures (4 to 40°C) implied a possible use in several organs in humans (acidity in the stomach and alkaline in the small intestine) that the body temperature can reach 40°C. Despite possible complex criteria of phage selection in the real clinical setting; for example, high coverage on bacterial strains without antibiotic resistant genes), vB_PaeM-AL (a lytic phage) was selected here just for a proof of concept on

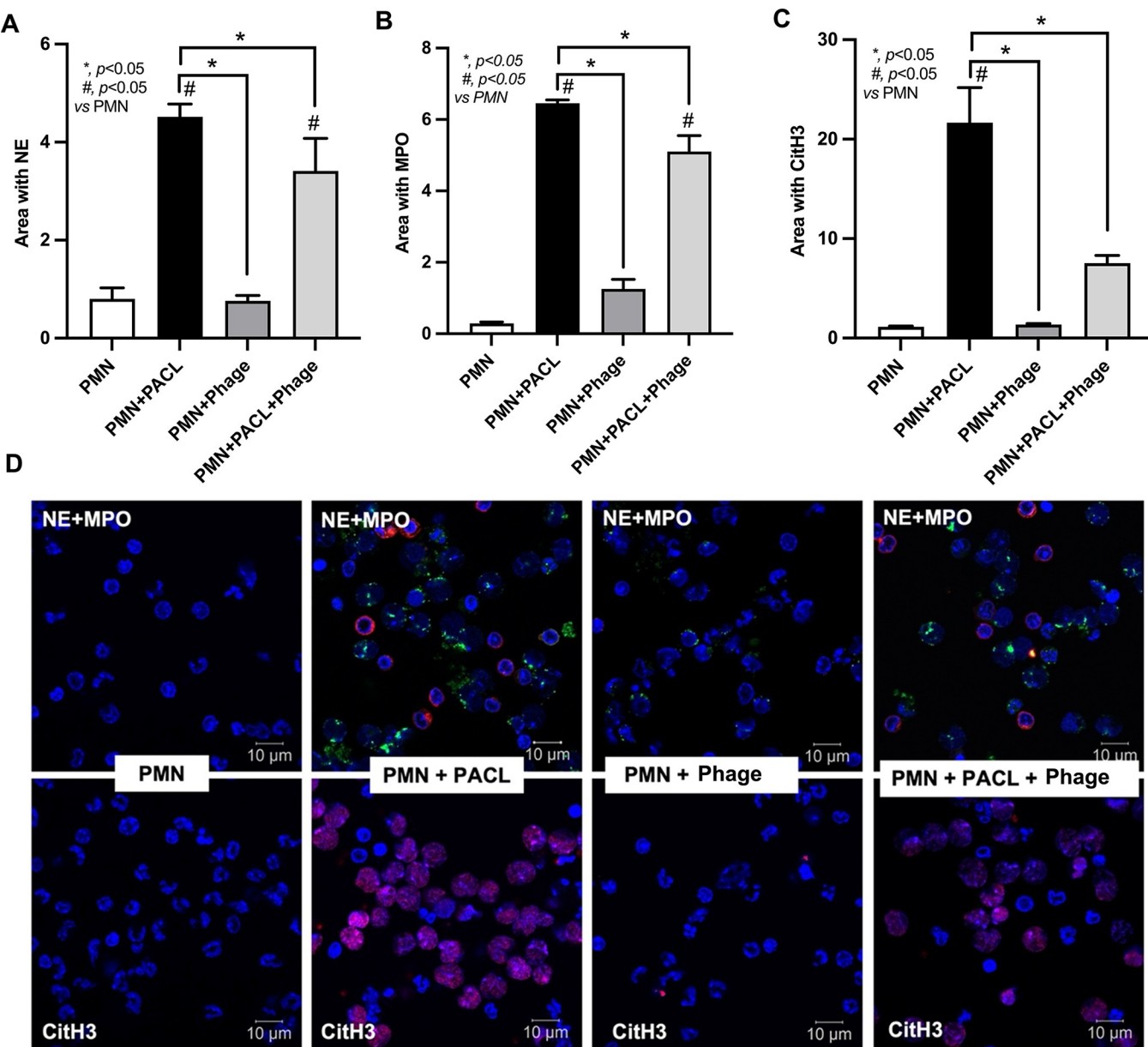

**Fig 6.** The effects of phage therapy on the neutrophil extracellular trap response as indicated by the immunofluorescence staining using neutrophil elastase (NE) (red), myeloperoxidase (MPO) (green), and citrullinated histone 3 (CitH3) (red) are indicated by the area with NE (**A**), MPO (**B**), CitH3 (**C**) and the representative images (**D**) are demonstrated. The results were combined from the isolated triplicated experiments. Data are shown as the mean ± SEM.

phage isolation using animal sources. More studies on phage selection or phage combination therapy are interesting.

Although the administration of phages through the intrathecal (IT) route was hypothesized to have a better result than intravenous (IV) administration in pneumonia due to the direct contact with the pathogens, both routes indicated a similar effectiveness, especially survival rate and serum cytokines. However, the IT route more effectively reduced bacteria in BALF than the IV route, while the IV route induced higher phage abundance in serum without an obvious systemic immune activation. The similar phage abundance in BALF between IV and

IT routes implied a good kinetic of IV phage injection that might directly move toward the sites with bacteria (the hosts of phages). Hence, both IT and IV phage administration can be applied in patients with pneumonia as the IT route for intubated cases and the IV injection for patients without ventilators. The bactericidal activity of bacteriophages was also documented here in both BALF and in blood consistent with previous works [33, 34]. Without bacterial administration, phages were detectable in BALF and in blood after IT and IV routes, respectively, at 24 h after phage injection indicating a practical duration of phages that existed after treatment. With bacterial administration, the abundance of bacteriophages through both IT and IV routes is increased supporting the phage self-replication [35]. The administration of phages in control non-pneumonia mice indicated the safety of phage treatment and the reduced inflammatory cytokines (TNF-α and IL-6) in the serum of pneumonia mice supported the downregulation of proinflammatory cytokines by phages [36]. The reduced NETs in the lung and serum IL-1β, an important cytokine for the formation of NETs [37], after phage treatment indicated an impact of the less abundant bacteria after phage treatment. Despite the reduced NETs formation by bacteriophages, the enhanced neutrophil bactericidal activity was demonstrated here supporting previous publications [15–18]. Our in vitro experiment indicated the neural effect of Pseudomonas phage vB_PaeM-AL alone (without bacteria) on NETs formation supports a well-known low immune activation property of phages [38]. Although phage treatment alone could reduce bacterial abundance, the presence of both neutrophils and phages demonstrated an additive effect on bactericidal activity (**Fig 5A**).

There are several limitations in our study. First, the sequencing analysis was not performed because the discovery of the new phage was not our primary objective. Our current concept focused on testing the phage isolation method from feces of the living animals and demonstrated the possibility of isolation. Second, other parameters of NETs were not measured. In part, this is because the bactericidal activity of our phages cannot be explained through the enhanced NETs. Third, the combination of several phages, a well-known strategic use of phages, was not tested. Nevertheless, we demonstrated a proof of concept to extract phages from mouse feces, implying that there are some phages against human pathogenic bacteria in mouse feces. For the future direction, rapid extraction and enhancement of the specific phages against bacteria, especially the antibiotic resistant strains, that are cultured from the patients through the administration of these bacteria into the mice might be able to use as a personalized medicine. Although a tremendous phage library and preparation pipeline for emergency phage therapy are mentioned to use as a personalized treatment against antibiotic resistant strains [39], laboratory tests to select the specific phage is still difficult, the newly extracted phages against the bacteria that are cultured from patient specimens using several methods, including mice, to accelerate the process will be another strategy of phage personalized medicine in the future. More studies are interesting.

In conclusion, the isolation of phages from feces might be a first step for the concept of bacteriophage isolation from the individual patient to treat that patient to avoid the necessity of a phage library. The international collaboration working on several aspects; for example, enhancing the possibility of phage discovery (specific mouse strains, bedding, and probiotics), rapid phage propagation methods, prevention of phage incorporation into the targeted bacteria, and product preparation for human use, might accelerate the progression of phage therapy. More studies are warranted.

## Supporting information

**S1 Fig. Study flowchart.**
(TIF)

**S1 Data.**
(XLSX)

## Acknowledgments

We thank Dr. Tanittha Chatsuwan, Department of Microbiology, Faculty of Medicine, Chulalongkorn University, for kind providing the left-over samples in the study.

## Author Contributions

**Conceptualization:** Nuttawut Sutnu, Wiwat Chancharoenthana, Supitcha Kamolratanakul, Asada Leelahavanichkul.

**Data curation:** Wiwat Chancharoenthana, Pornpimol Phuengmaung, Uthaibhorn Singkham-In, Chiratchaya Chongrak, Sirikan Montathip, Tanittha Chatsuwan.

**Formal analysis:** Nuttawut Sutnu, Wiwat Chancharoenthana, Supitcha Kamolratanakul, Chiratchaya Chongrak, Sirikan Montathip, Dhammika Leshan Wannigama, Tanittha Chatsuwan, Asada Leelahavanichkul.

**Funding acquisition:** Nuttawut Sutnu, Wiwat Chancharoenthana, Asada Leelahavanichkul.

**Investigation:** Nuttawut Sutnu, Wiwat Chancharoenthana, Pornpimol Phuengmaung, Uthaibhorn Singkham-In, Chiratchaya Chongrak, Sirikan Montathip, Dhammika Leshan Wannigama, Tanittha Chatsuwan, Asada Leelahavanichkul.

**Methodology:** Wiwat Chancharoenthana, Supitcha Kamolratanakul, Pornpimol Phuengmaung, Uthaibhorn Singkham-In, Chiratchaya Chongrak, Sirikan Montathip, Dhammika Leshan Wannigama, Tanittha Chatsuwan, Asada Leelahavanichkul.

**Project administration:** Nuttawut Sutnu.

**Resources:** Pornpimol Phuengmaung, Uthaibhorn Singkham-In, Chiratchaya Chongrak, Dhammika Leshan Wannigama, Tanittha Chatsuwan, Puey Ounjai.

**Software:** Nuttawut Sutnu, Pornpimol Phuengmaung, Uthaibhorn Singkham-In, Dhammika Leshan Wannigama, Tanittha Chatsuwan, Puey Ounjai.

**Supervision:** Dhammika Leshan Wannigama, Puey Ounjai, Marcus J. Schultz, Asada Leelahavanichkul.

**Validation:** Nuttawut Sutnu, Wiwat Chancharoenthana, Dhammika Leshan Wannigama, Puey Ounjai, Asada Leelahavanichkul.

**Visualization:** Dhammika Leshan Wannigama, Puey Ounjai, Marcus J. Schultz, Asada Leelahavanichkul.

**Writing – original draft:** Nuttawut Sutnu, Wiwat Chancharoenthana, Supitcha Kamolratanakul, Marcus J. Schultz, Asada Leelahavanichkul.

**Writing – review & editing:** Wiwat Chancharoenthana, Marcus J. Schultz, Asada Leelahavanichkul.

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
