## [Decision Letter · Decision Letter 0]

31 May 2024

PONE-D-24-09029Bacteriophages isolated from mouse feces attenuates pneumonia mice caused by Pseudomonas aeruginosaPLOS ONE

Dear Dr. Chancharoenthana,

Thank you for submitting your manuscript to PLOS ONE. After careful consideration, we feel that it has merit but does not fully meet PLOS ONE’s publication criteria as it currently stands. Therefore, we invite you to submit a revised version of the manuscript that addresses the points raised during the review process. Your manuscript has been reviewed by an expert in this field.  However, a minor revision is suggested  and I agree.  Please follow the comments and make the necessary revision. 

We look forward to receiving your revised manuscript.

Kind regards,

Yung-Fu Chang

Academic Editor

PLOS ONE

Journal Requirements:

2. To comply with PLOS ONE submissions requirements, in your Methods section, please provide additional information regarding the experiments involving animals and ensure you have included details on (1) methods of sacrifice, (2) efforts to alleviate suffering.

3. For studies reporting research involving human participants, PLOS ONE requires authors to confirm that this specific study was reviewed and approved by an institutional review board (ethics committee) before the study began. Please provide the specific name of the ethics committee/IRB that approved your study, or explain why you did not seek approval in this case.

4. Please provide additional details regarding participant consent. In the ethics statement in the Methods and online submission information, please ensure that you have specified what type you obtained (for instance, written or verbal, and if verbal, how it was documented and witnessed). If your study included minors, state whether you obtained consent from parents or guardians. If the need for consent was waived by the ethics committee, please include this information.

6. Thank you for stating the following financial disclosure: 

"This research project is supported by the NSRF via the Program Management Unit for Human Resources & Institutional Development, Research and Innovation (B16F640175) and (B36G660003) and (B48G660112), Rachadapisek Sompote Matching Fund (RA-MF-22/65 and RA-MF-13/66), the National Research Council of Thailand (NRCT-N41A640076 and NRCT-N34A660583), and Fundamental Fund 2567. NS is funded by the 90th Anniversary of Chulalongkorn University Fund (Ratchadapisek Sompote Endowment Fund). WC is funded by Mahidol University (Fundamental Fund; Basic Research Fund: fiscal year 2022)."

7. We note that your Data Availability Statement is currently as follows: All relevant data are within the manuscript and its Supporting Information files.

Reviewers' comments:

Reviewer's Responses to Questions

**Comments to the Author**

1. Is the manuscript technically sound, and do the data support the conclusions?

Reviewer #1: Yes

2. Has the statistical analysis been performed appropriately and rigorously? 

Reviewer #1: Yes

3. Have the authors made all data underlying the findings in their manuscript fully available?

Reviewer #1: Yes

4. Is the manuscript presented in an intelligible fashion and written in standard English?

Reviewer #1: Yes

5. Review Comments to the Author

Reviewer #1: The manuscript offers an in-depth investigation into the isolation and characterization of bacteriophages from mouse feces and their ability to treat Pseudomonas aeruginosa pneumonia. The observation is methodologically sound and explores a novel technique that could decorate the remedy of bacterial infections, mainly in the context of growing antibiotic resistance. However, several critical factors require attention before it is considered for publication.

Introduction:

- The usage of phages isolated from the mice and how helpful they are as compared with the other animal or environmental sources can be cited to highlight its importance. Also, it would be helpful to intricate on why this technique is predicted to be advanced to present strategies.

- The mortality records cited are useful but may be impactful if connected directly to the need for alternative therapies like bacteriophage therapy against antibiotics or others. Providing more inputs from a broader geographical context may also be relevant.

Methods:

- The strategies used from the isolation, and characterization, to applications/treatment, can be summarized in a flow chat to have an overall understanding of the study conducted.

- The criteria for selecting the specific bacterial strains (PA1, PA2, and many others.) as well as phages ought to be clarified. Were these cell lines chosen totally on their clinical relevance, resistance profiles, or any other particular attributes? Also, the phage selection criteria must be explained in detail.

- The phage purification using cesium chloride density gradient must be explained in detail.

Results:

- The effects of isolating phages from mouse feces and the effectiveness of those phages in treating pneumonia have to be explicitly stated.

- The effect of isolated phages on different host stains and their lysis can be better presented with the inclusion of additional comparative analyses, possibly through graphical representations like bar charts or heatmaps. To demonstrate its significance.

- All the studies have been properly conducted but the significance of the animal model, histology, as well as in-vitro studies with neutrophils must be explained in detail with proper justification for their findings! To provide further importance to the observed results.

Discussion:

- The specificity of phages isolated from feces is well reported, but potential solutions or future directions to such studies concerning effective solutions of antibiotic-resistant strains must be explained with limitations.

- The ability to translate these findings to human clinical settings has to be discussed with possible hurdles and potential future studies.

Conclusion:

- A call to action for additional research or collaboration may also add significance to the manuscript.

6. PLOS authors have the option to publish the peer review history of their article (what does this mean?). If published, this will include your full peer review and any attached files.

Reviewer #1: **Yes: **Dr. Swapnil Ganesh Sanmukh

---

## [Author Response · Author response to Decision Letter 0]

14 Jun 2024

Response to the Reviewers

Reviewer #1: The manuscript offers an in-depth investigation into the isolation and characterization of bacteriophages from mouse feces and their ability to treat Pseudomonas aeruginosa pneumonia. The observation is methodologically sound and explores a novel technique that could decorate the remedy of bacterial infections, mainly in the context of growing antibiotic resistance. However, several critical factors require attention before it is considered for publication.

Introduction:

- The usage of phages isolated from the mice and how helpful they are as compared with the other animal or environmental sources can be cited to highlight its importance. 

ANS: We appreciate the reviewer's feedback and would like to share our thoughts as follows: "We chose mouse feces for the proof of concept because i) humans are mammals and phages isolated from mammals may be more similar to their use in humans, and ii) experimental mice are small enough to use as an initial recovery experiment and future experiments for improved phage isolation in mice are feasible and easier than the larger mammals."

Also, it would be helpful to intricate on why this technique is predicted to be advanced to present strategies.

ANS: We appreciate the reviewer's feedback and would like to share some of our ideas: "Since phage isolation doesn't necessitate a significant amount of feces, fecal collection and other manipulations on mice are more straightforward than in larger animals; for instance, administering certain molecules that could potentially enhance the target phage. Additionally, phages targeting specific bacteria may exhibit greater specificity in certain mouse strains that have undergone genetic modifications, a prospect for future research. "

- The mortality records cited are useful but may be impactful if connected directly to the need for alternative therapies like bacteriophage therapy against antibiotics or others. Providing more inputs from a broader geographical context may also be relevant.

ANS: We appreciate the reviewer's feedback and have included a sentence in the introduction that highlights the potential of using phages as an alternative treatment as “Therefore, we suggest that bacteriophage therapy could serve as an alternative therapy or an adjuvant strategy to combat bacterial infections.”

Methods:

- The strategies used from the isolation, and characterization, to applications/treatment, can be summarized in a flow chat to have an overall understanding of the study conducted.

ANS: We thank the reviewer for their comments and have added a flow chart to the method section.

- The criteria for selecting the specific bacterial strains (PA1, PA2, and many others.) as well as phages ought to be clarified. Were these cell lines chosen totally on their clinical relevance, resistance profiles, or any other particular attributes? Also, the phage selection criteria must be explained in detail.

ANS: We thank the reviewer for comments. We just use all bacterial strain that are available in our lab and the best phage that cover the highest number of PA. Because we just try to prove the concept of phase isolation from samples of living animal. Then, only the simple criteria were used. Then, we added in the method section as following; “As such, the isolated stains of all currently available bacteria, including”, and added in the result section as following “Different from the isolation from environmental sources, there were only 4 different bacteriophages from our results (Table 1) and pseudomonas phage vB_PaeM-AL was selected for further test in vivo and in vitro due to the coverage in most of the P. aeruginosa strains used in our experiments. Also, the table of host range determination is expanded to other phages that were not selected to test (a new table 1) and some sentences are also added in the new discussion as following; “Despite possible complex criteria of phage selection in the real clinical setting; for example, high coverage on bacterial strains without antibiotic resistant genes), vB_PaeM-AL (a lytic phage) was selected here just for a proof of concept on phage isolation using animal sources. More studies on phage selection or phage combination therapy are interesting.”. 

- The phage purification using cesium chloride density gradient must be explained in detail.

ANS: We thank the reviewer for comments and add more details in the new method section. 

Results:

- The effects of isolating phages from mouse feces and the effectiveness of those phages in treating pneumonia have to be explicitly stated.

ANS: We thank the reviewer for comments. Due to the limitation of animal study protocol, we can teste in mice only with a single phage, despite a good matching between in vitro and in vivo. Then, we discuss the possibly correlation between the in vitro and in vivo and the lack of test in other non-selected phages as a limitation of our study. 

- The effect of isolated phages on different host stains and their lysis can be better presented with the inclusion of additional comparative analyses, possibly through graphical representations like bar charts or heatmaps. To demonstrate its significance.

ANS: We apologize for the unclear presentation. We use only PACL, a strain that previously successfully activate diseases in mice, and the selected phage is an only strain that can neutralize PACL (as shown in the new table1). Hence, we more clearly mentioned the use of PACL in the new result section as following; “Although Pseudomonas phage vB_PaeM-AL can neutralize several pseudomonas strains (Table 1), PACL, a strain with previously demonstrated pathogenicity in an animal model (wound model) [6], might also be able to induce mouse pneumonia was selected to use for further experiment”. 

- All the studies have been properly conducted but the significance of the animal model, histology, as well as in-vitro studies with neutrophils must be explained in detail with proper justification for their findings! To provide further importance to the observed results.

ANS: We thank the reviewer for the comment and add more explanation in the new result section as following “Without phage treatment, the lethal pneumonia of our model was indicated by approximately 90% mortality rate within 7 days after bacterial administration (Fig 2A) with high bacterial abundance (BALF and in blood), elevated inflammatory cytokines (Fig 2B - I), and lung consolidation (the presence of exudate in the airways and alveoli) (Fig 3A, B), which was compatible with severe lobar pneumonia in patients.” for our pneumonia model. For the in vitro findings, we rewrite all of the result of this section and conclude as following; “Although bacterial abundance between bacteria + phage + PMN and bacteria + phage was similar (Fig 5A), lower phages in the latter group was required for this similar reaction (Fig 5B) and PMN could recognize phages as indicated by increased IL-6 (Fig 5D on phage + PMN group), but not NET activation (Fig 6A-D). Thus, bactericidal activity of our phage might, at least in part, be through the activation of neutrophils (cytokine production but not NETs). 

Discussion:

- The specificity of phages isolated from feces is well reported, but potential solutions or future directions to such studies concerning effective solutions of antibiotic-resistant strains must be explained with limitations.

ANS: We thank the reviewer for the comment and add more discussion as following; “Although a tremendous phage library and preparation pipeline for emergency phage therapy are mentioned to use as a personalized treatment against antibiotic resistant strains [39], laboratory tests to select the specific phage is still difficult, the newly extracted phages against the bacteria that are cultured from patient specimens using several methods, including mice, to accelerate the process will be another strategy of phage personalized medicine in the future. More studies are interesting.”.

- The ability to translate these findings to human clinical settings has to be discussed with possible hurdles and potential future studies.

ANS: We thank the reviewer for the comment and add more discussion as following; “Nevertheless, we demonstrated a proof of concept to extract phages from mouse feces, implying that there are some phages against human pathogenic bacteria in mouse feces. For the future direction, rapid extraction and enhancement of the specific phages against bacteria, especially the antibiotic resistant strains, that are cultured from the patients through the administration of these bacteria into the mice might be able to use as a personalized medicine.”.

Conclusion:

- A call to action for additional research or collaboration may also add significance to the manuscript.

ANS: We thank the reviewer for the comment and add a few sentences as following; “The international collaboration working on several aspects; for example, enhancing the possibility of phage discovery (specific mouse strains, bedding, and probiotics), rapid phage propagation methods, prevention of phage incorporation into the targeted bacteria, and product preparation for human use, might accelerate the progression of phage therapy. More studies are warranted.”

---

## [Decision Letter · Decision Letter 1]

1 Jul 2024

Bacteriophages isolated from mouse feces attenuates pneumonia mice caused by Pseudomonas aeruginosa

PONE-D-24-09029R1

Dear Dr. Chancharoenthana,

We’re pleased to inform you that your manuscript has been judged scientifically suitable for publication and will be formally accepted for publication once it meets all outstanding technical requirements.

Kind regards,

Yung-Fu Chang

Academic Editor

PLOS ONE

Additional Editor Comments (optional):

Reviewers' comments:

Reviewer's Responses to Questions

**Comments to the Author**

1. If the authors have adequately addressed your comments raised in a previous round of review and you feel that this manuscript is now acceptable for publication, you may indicate that here to bypass the “Comments to the Author” section, enter your conflict of interest statement in the “Confidential to Editor” section, and submit your "Accept" recommendation.

Reviewer #1: All comments have been addressed

2. Is the manuscript technically sound, and do the data support the conclusions?

Reviewer #1: Yes

3. Has the statistical analysis been performed appropriately and rigorously? 

Reviewer #1: Yes

4. Have the authors made all data underlying the findings in their manuscript fully available?

Reviewer #1: Yes

5. Is the manuscript presented in an intelligible fashion and written in standard English?

Reviewer #1: Yes

6. Review Comments to the Author

Reviewer #1: The author's have addressed most of the comments and the arricle is significantly improved from its previous version.

7. PLOS authors have the option to publish the peer review history of their article (what does this mean?). If published, this will include your full peer review and any attached files.

Reviewer #1: **Yes: **Swapnil Ganesh Sanmukh

---

## [Editor Report · Acceptance letter]

5 Jul 2024

PONE-D-24-09029R1 

PLOS ONE

Dear Dr. Chancharoenthana, 

I'm pleased to inform you that your manuscript has been deemed suitable for publication in PLOS ONE. Congratulations! Your manuscript is now being handed over to our production team.

Kind regards, 

on behalf of

Dr. Yung-Fu Chang 

Academic Editor

PLOS ONE